# Photonic Weyl degeneracies in magnetized plasma

Wenlong Gao[1,2], Biao Yang[1], Mark Lawrence[1,3], Fengzhou Fang[2], Benjamin Béri[1] & Shuang Zhang[1,2]

Weyl particles are elusive relativistic fermionic particles with vanishing mass. While not having been found as an elementary particle, they are found to emerge in solid-state materials where three-dimensional bands develop a topologically protected point-like crossing, a so-called Weyl point. Photonic Weyl points have been recently realised in three-dimensional photonic crystals with complex structures. Here we report the presence of a novel type of plasmonic Weyl points in a naturally existing medium—magnetized plasma, in which Weyl points arise as crossings between purely longitudinal plasma modes and transverse helical propagating modes. These photonic Weyl points are right at the critical transition between a Weyl point with the traditional closed finite equifrequency surfaces and the newly proposed 'type II' Weyl points with open equifrequency surfaces. Striking observable features of plasmon Weyl points include a half $k$-plane chirality manifested in electromagnetic reflection. Our study introduces Weyl physics into homogeneous photonic media, which could pave way for realizing new topological photonic devices.

[1] School of Physics and Astronomy, University of Birmingham, Birmingham, B15 2TT, UK. [2] State Key Laboratory of Precision Measuring Technology and Instruments, Tianjin University, Tianjin, 300072, China. [3] Department of Materials Science and Engineering, Stanford University, Stanford, California, 94305, USA. Correspondence and requests for materials should be addressed to B.B. (email: b.beri@bham.ac.uk) or to S.Z. (email: s.zhang@bham.ac.uk).

In recent years, studies of photonic topological properties aroused a significant amount of research attention. On one aspect, periodic structures like photonic crystals[1–4], waveguides[5–7], optical resonators[8–10], metamaterials[11–13] and polaritons[14,15] are studied on their ability to support topological non-trivial edge states mimicking electrons in condensed matter theory. Previous realizations are mainly in two-dimensional photonic systems, where Dirac points[16,17] (or quadratic degeneracies, which consist of two Dirac points[17]) found at high-symmetry points can be described by the Hamiltonian $H = v_x k_x \sigma_x + v_y k_y \sigma_y$ ($\sigma_{x,y,z}$ are Pauli matrices). Opening such Dirac points by breaking time reversal symmetry[1], introducing spin–orbital coupling[6,11] or synthetic gauge fields[8,9], one can attain gaps with non-zero Chern numbers or spin Chern numbers that could lead to topologically protected edge states.

The possibility of such gap openings shows that the two-dimensional Dirac points and the corresponding massless excitations are unstable against perturbations[18]. The situation drastically changes in three-dimensional: so-called Weyl points, governed by the Weyl Hamiltonian $H = v_x k_x \sigma_x + v_y k_y \sigma_y + v_z k_z \sigma_z$, provide massless modes that are highly robust[19–22]. A direct way to see this is to note that the inclusion of all three Pauli matrices exhausts all degrees of freedom allowed for a two-band description, making such Weyl-type band crossing protected. More abstractly, Weyl points can be seen as momentum space monopoles of quantized monopole charge, acting as sources of Berry curvature, with the sign of the monopole charge set by the Weyl point's chirality[19]. Within this viewpoint, the stability is understood by noting that the only way to eliminate Weyl points is the annihilation of two anti-chirality Weyl points together. Recently, systems supporting such Weyl points, so-called Weyl semimetals, were proposed in electronic[23–26], photonic[27] and acoustic systems[28], which could potentially start a new era in realizing topological states. So far, in the photonic context, Weyl points have been realized in photonic crystals with precisely engineered lattice structures, periodic on the scale of the wavelength[22].

Here we discover a new type of Weyl points in a homogeneous photonic material: free electron gas under static bias magnetic field (or magnetized plasma). With strong enough applied magnetic field, the cyclotron frequency exceeds the plasma frequency, which results in crossings between the longitudinal plasmon mode and the helical propagating mode at the plasma frequency. These crossing points in the momentum space serve as Weyl points that are responsible for all the non-vanishing Berry curvature and non-trivial topological features in this system. Importantly, magnetized plasma exhibits parabolic equifrequency surfaces (EFSs, the photonic analogue of Fermi surfaces) near the Weyl points, which to our knowledge, has not yet found a counterpart in condensed matter systems. We also predict salient observable features of these Weyl points in the reflection spectra, including a peculiar polarization pattern carrying fingerprints of the 'eigen-state' properties near the Weyl points. The homogeneity of the system greatly facilitates the investigation of various interesting physics associated with the Weyl degeneracies without relying on complex numerical simulations. In addition, the magnetized plasma approach proposed here does not involve complicated structural design and fabrication, and the system can be reconfigured in real time by varying the plasma density and the strength or direction of the applied static magnetic field.

## Results

### Hamiltonian formalism of magnetized plasma.

Unlike artificial metamaterials that could possess highly non-local response caused by either non-local modes in 'meta-atoms' or Brillouin zone boundary, the cold magnetized plasma's optical response can be considered to be local as long as the wavelength of interest is far greater than the mean spacing of the charged particles in the plasma. To start, we develop the Hamiltonian formalism of magnetized plasma based on a previous work by Raman et al.[29] (see Supplementary Note 1 for detailed derivation):

$$\omega_p \begin{bmatrix} 0 & -\mathbf{K}\times/k_p & -i \\ \mathbf{K}\times/k_p & 0 & 0 \\ i & 0 & (\omega_c\Delta - i\Gamma I)/\omega_p \end{bmatrix} \begin{bmatrix} \mathbf{E} \\ \mathbf{H} \\ \mathbf{V} \end{bmatrix} = \omega \begin{bmatrix} \mathbf{E} \\ \mathbf{H} \\ \mathbf{V} \end{bmatrix} \quad (1)$$

where $\omega_p = Ne^2/m$ is plasma frequency, $\omega_c = eB/m$ is cyclotron frequency, $\mathbf{V}$ is the electron's velocity field, $k_p = \frac{\omega_p}{c}$. The damping frequency $\Gamma$ can be neglected as it can be orders of magnitude less than the plasma frequency in a gaseous plasma[30]. Each element in the matrix is a 3-by-3 matrix, $c$ is the speed of light in vacuum, $I$ is unit matrix, and the 3-by-3 matrix $\Delta = [\sigma_y, 0; 0, 1]$ represents the coupling between $V_x$ and $V_y$ induced by Lorentz force. In realistic cold plasma, presence of ions can also be formulated into the Hamiltonian, which results in nearly negligible effect to the dispersion close to the electron plasma frequency (Supplementary Fig. 1).

Dimension of the Hamiltonian in equation (1) is 9-by-9, which results in an overall of nine dispersion curves—four dispersion curves in the positive frequency regime, four dispersions in the negative frequency regime and one zero frequency mode. Since the dispersion curves in the positive and negative frequencies are linked to each other through the transformation: $\omega \to -\omega$ and $[\mathbf{E}, \mathbf{H}, \mathbf{V}]^T \to [\mathbf{E}^\star, -\mathbf{H}^\star, \mathbf{V}^\star]^T$, we therefore only need to consider the dispersion curves in the positive frequency regime, which are calculated for $\omega_c = 1.2\omega_p$ and plotted in Fig. 1a. Band structures obtained in the range $\omega_c < \omega_p$ is given in Supplementary Fig. 2.

### Weyl points.

Across the energy bands, there is a straight line along the $K_z$ axis at $\omega_c = \omega_p$. This is the longitudinal bulk plasmon mode occurring at $\varepsilon_z = 0$, which is induced by the Drude dispersion. It is clearly shown that there are four linear degeneracies along the $K_z$ axis: two between the first and the second bands, other two between the second and the third bands. The linear degeneracies are between the bulk plasmon and the circular polarized propagating modes, and are protected, in physical terms, by the orthogonality of the polarizations. Mathematically, as will be made clear later, these linear degeneracies are Weyl points and as such are sources and drains of Berry curvature flux lines. The location of the linear degeneracies in momentum space is expressed by

$$K_z^{weyl} = \pm\sqrt{\frac{\omega_c}{\omega_c \pm \omega_p}} \quad (2)$$

which is plotted in Fig. 1b. Note that the outer Weyl points go to infinity for $\omega_c = \omega_p$, and leads to a transition of Berry curvature in infinity. The detailed analysis of this asymptotic behaviour of the Berry curvature is discussed in Supplementary Note 2 and Supplementary Figs 3 and 4. For $\omega_c < \omega_p$, only the inner Weyl points (black curve in Fig. 1b) are present.

Using the Hamiltonian in equation (1), the Berry curvature of band $n$ can be computed by $\Omega_n = i \sum_{m \neq n} (\langle n|\partial_{R_1}|m\rangle\langle m|\partial_{R_2}|n\rangle - (R_1 \leftrightarrow R_2))/(E_n - E_m)^2$ (refs 31,32). It is crucial to stress that the Berry curvatures satisfy spatial inversion symmetry: $\Omega(\mathbf{K}) = \Omega(-\mathbf{K})$[33] (derivation given in Supplementary Note 3) and thus the chirality of the Weyl points satisfies the relation $\gamma(K_z) = -\gamma(-K_z)$. Because of this symmetry of the Berry curvature, any closed EFS centred at the origin attain zero total Berry flux, or Chern number. Therefore, we are more interested in the open hyperbolic EFSs embedded in the first band, and thus

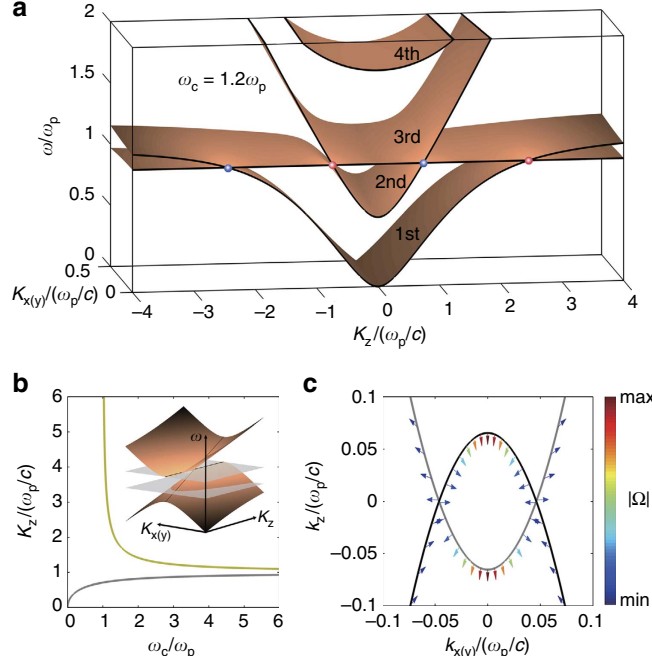

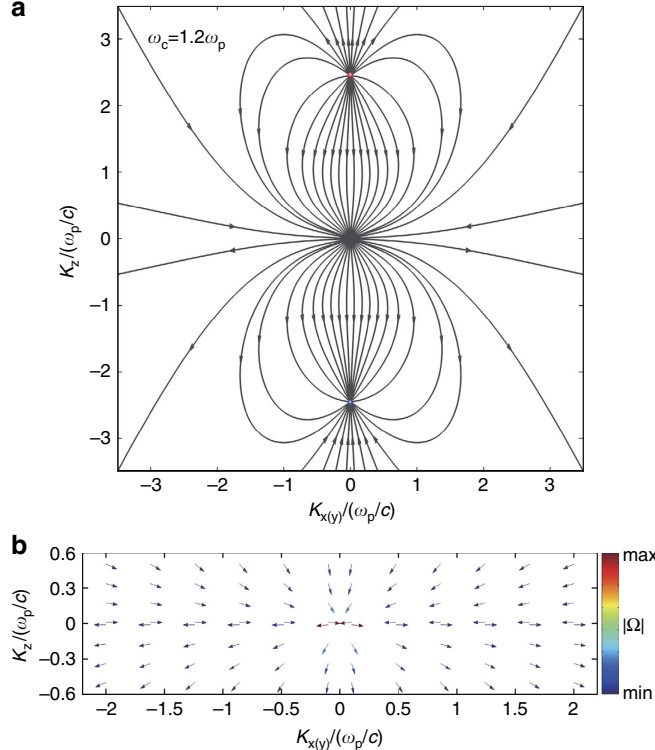

**Figure 1 | Dispersion relation of a magnetized plasma.** (a) Three-dimensional energy band of the magnetized plasma for $\omega_c = 1.2\omega_p$, 'Weyl degeneracies' are highlighted by the coloured spots (red for $+1$ and blue for $-1$). Each slice of this band structure at some definite $k_z$ is also a cross-section plot of an arbitrary plane across the momentum space's centre in $x$–$y$ plane. The straight line at $\omega = \omega_p$, $K_x = K_y = 0$ is the longitudinal plasma mode. The bands are numbered from bottom to top, and the fourth band is not shown here for conciseness. (b) Plot of equation (7) in one quadrant of the parameter space. The outer Weyl points (yellow) goes to infinity when $\omega_c = \omega_p$, while the inner Weyl points (grey) annihilate at the origin. The inset shows the energy bands of the effective Hamiltonian of the plasmon Weyl points. Intersection between the grey planes that pin a definite frequency, and the energy band is a straight line at $\omega = \omega_p$, while is parabola at shifted frequencies. (c) 'Fermi surfaces' and Berry curvature plot when shifted frequency $\delta\omega$ equals $-0.01$(grey, corresponds to the first band) and $0.01$(black, corresponds to the second band), respectively. The colours indicate normalized intensity of Berry curvatures.

the outer Weyl points between first and second bands. The Berry curvature vector of this band is plotted in Fig. 2a, showing a source (marked red) and a drain (marked blue) of Berry curvatures in accordance with the fact that chirality $\gamma$ is 1 ($-1$) for positive (negative) $K_z$. Interestingly, the Berry flux lines are concentrated at the origin of the momentum space. Because of the symmetry relation: $\Omega(\mathbf{K}) = \Omega(-\mathbf{K})$, the origin functions as a source in the upper half $k$-space and a drain in the lower half $k$-space.

To better understand these linear degeneracies, we apply $k \cdot p$ theory[16] (effective Hamiltonian theory) to obtain the approximate Hamiltonian near the degeneracies (the details can be found in Supplementary Note 4). Expanding to first order in the vicinity of the outer degeneracy, we find the effective Hamiltonian

$$H = Nk_x\sigma_x + \boldsymbol{\sigma}_s \cdot \mathbf{B}_s Nk_y\sigma_y + \mathbf{S} \cdot \mathbf{B}_s\left(\frac{M}{2}k_z\sigma_z + \frac{M}{2}k_zI\right) \quad (3)$$

Where $\mathbf{B}_s$ is a unit vector along the direction of the static magnetic field, and $\mathbf{S}$ is a unit vector along the Poynting vector of the helical propagating mode. $\boldsymbol{\sigma}_s$ is the spin vector of the helical

**Figure 2 | Berry flux distribution of the first band in magnetized plasma.** (a) Magnetic field line plot of Berry curvature of the first band. Weyl points are coloured red (blue) for positive (negative) chirality. (b) Plot of Berry curvatures around the $K_x - K_y$ plane. Colours represent intensity of Berry curvatures normalized to the maximum, and arrows indicate directions. As is shown, in vicinity to the plane, Berry curvatures are parallel to it.

propagating mode defined by $\boldsymbol{\sigma}_s = 2(\text{Re}\mathbf{E} \times \text{Im}\mathbf{E})/|\mathbf{E}|^2$ (ref. 34). Here the coordinate origin is shifted to the Weyl point, and $\mathbf{k} = [k_x, k_y, k_z]^T$ represents small deviations from the degeneracy in momentum space. $M$ and $N$ are real and can be expressed in terms of $\omega_c$ and $\omega_p$ (Supplementary Note 4). Equation (3) is very similar to the expression of Weyl dispersion except for the last term. It is worth noticing that when $\delta k_x$ and $\delta k_y$ are zero, the eigenvalues of this Hamiltonian are simply 0 and $Mk_z$. The eigenvalue 0 corresponds to the longitudinal plasma mode, which is a straight line at $\omega = \omega_p$. Interestingly, as is shown in the inset of Fig. 1b, this Weyl Hamiltonian represents a tilted Weyl cone[30] and is identified to be at the transition between type I Weyl points with a spherical or, more generally, ellipsoid EFSs, and type II Weyl points with hyperbolic EFSs[35]. Indeed, at frequencies slightly shifted away from the 'Weyl point frequency', the Hamiltonian given by equation (3) results in highly anisotropic parabolic EFSs (Fig. 1c). Chirality for the Weyl point is defined as $\gamma = sgn(v_xv_yv_z)$[19–21,36,37], thus is $sgn(\boldsymbol{\sigma}_s \cdot \mathbf{B}_s\mathbf{S} \cdot \mathbf{B}_s)$ according to equation (3). Berry curvature from this type of Weyl point can be readily expressed by (Supplementary Note 5):

$$\Omega(\lambda_{\text{down}}) = -\Omega(\lambda_{\text{up}}) = \gamma\frac{\alpha}{2\left(\alpha^2k_z^2 + k_r^2\right)^{3/2}}\mathbf{k} \quad (4)$$

where $\lambda_{\text{up}}$ and $\lambda_{\text{down}}$ stands for upper band and lower bands (Fig. 2c) in the effective Hamiltonian, while $\alpha = \frac{M}{2N}$ and $k_r = \sqrt{k_x^2 + k_y^2}$. Equation (4) represents the Berry flux of a monopole located in an anisotropic $k$-space with effective permeability given by $[1, 1, \alpha^{-2}]$. Integrating the Berry curvature in equation (4) on the EFSs, one can obtain a value

of $\pm 2\pi$ (Supplementary Note 5), which corresponds to quantized Chern number of $\pm 1$. Intuitively, slope of the parabola is $2N^2 k_r/\omega M$ and becomes parallel to the $z$ axis at large momentum, thus collecting all the Berry curvature fields. Thus, the effective model not only proves the quantized gauge flux emitted from the Weyl points but also represents a perfect example that an open 'Fermi surface' could attain quantized Chern number. The chirality $\gamma$ of the outer Weyl point on the positive $k_z$ axis is equal to 1 when $\omega_c > 0$. As an example, the Berry curvatures on parabolic EFSs at $\omega_c = 1.2$ are shown in Fig. 1c; in this case $\alpha = \sqrt{2}/4$, which leads to an anisotropic gauge flux distribution. In the vicinity of the Weyl point, the parabolic EFS is an approximation to the hyperbolic bands that we are interested in. However, at very large $\mathbf{k}$, the parabolic EFS deviates significantly from the actual hyperbolic equi-frequency surface, whose slope at large $\mathbf{k}$ approaches their asymptotes expressed by

$$\sqrt{\frac{\omega_c^2-\omega^2}{\omega^2}\frac{\omega_p^2-\omega^2}{\omega_c^2+\omega_p^2-\omega^2}}.$$ It thus provides a possible channel through

which Berry curvatures of Weyl points can leak into the infinity. This results a non-integer integral of the Berry curvature over the hyperbolic EFS. Nonetheless, it is shown in Supplementary Figs 5 and 6 that this does not affect the presence of a Fermi arc that connects between the two hyperbolic EFSs. Finally, when loss is at present in the system, the Weyl point would split into two exceptional points that together contribute the same Berry phase as before (Supplementary Note 6 and Supplementary Fig. 7).

Close to the origin of the $k$-space, by applying $k \cdot p$ theory again we can construct a 3-by-3 effective Hamiltonian (containing the first band, its mirror image band in negative frequency regime and a trivial zero energy band), which is expressed as $-i\omega_c K_z \mathbf{K} \times$. It is straightforward to work out the analytical expression of Berry curvature of the first band around the origin as (Supplementary Note 7)

$$\begin{cases} \Omega_{\mathbf{k}} = -\dfrac{\mathbf{K}}{|\mathbf{K}|^3} \ (K_z > 0) \\ \Omega_{\mathbf{k}} = \dfrac{\mathbf{K}}{|\mathbf{K}|^3} \ (K_z < 0) \end{cases} \tag{5}$$

Equation (5) shows that the Berry curvature flux passing through the origin is quantized to unity, which is reminiscent of magnetic flux lines near an infinitesimal vortex in a superconductor. Further numerical calculation confirms that Berry curvature flux lines at $K_z = 0$ are parallel to the $K_x - K_y$ plane (Fig. 2c), which satisfies the boundary condition of magnetic field at a superconductor surface. Thus, an amusing viewpoint of the $K_x - K_y$ plane is that of an infinitely thin 'momentum space superconductor' with an infinitesimally small vortex lying at the origin. The discontinuity of the Berry flux lines across the $K_x - K_y$ plane indicates the presence of effective 'surface currents' on this plane.

**Observable features.** We now move to discussing what the observable consequences of these plasmon Weyl points are. One of the most direct probes is provided by optical detection through an angle-resolved reflection experiment[27]. In Fig. 3a we show the schematic diagram of the experiment, where a high refractive index material with $n_s = 4$ is attached to the magnetized plasma so that the interface is parallel to the applied magnetic field. We consider transverse electric (TE) polarized light shone on the interface with polar angle $\Theta$ and azimuthal angle $\psi$ (positive when it lies between positive $x$ and $y$ axes). The location of the outer Weyl point in momentum space could be detected from the reflection spectrum. As is shown in Fig. 3a, at $\psi = 0$, the incident plane contains the projection of the Weyl point (red spot). A linear degeneracy is clearly observed as the touching point between two bands at $\omega = \omega_p$ in the reflectance spectrum (Fig. 3b). However when $\psi$ is not zero, a gap opens up

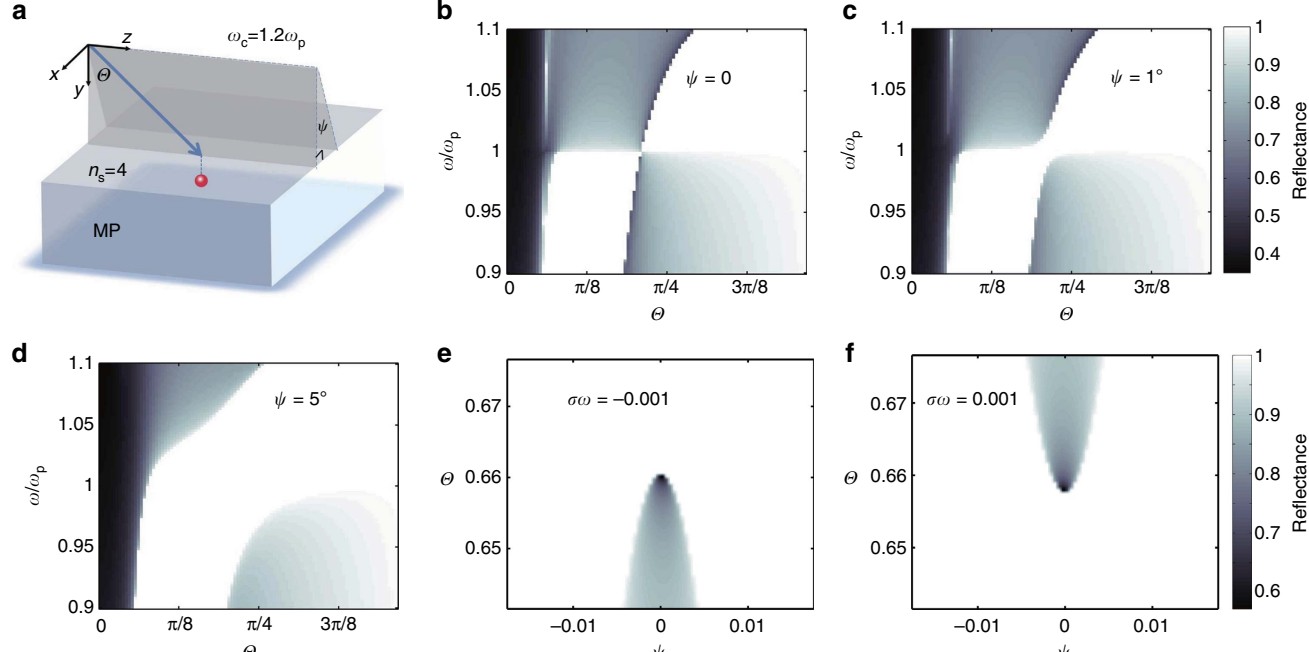

**Figure 3 | Weyl points probed by reflection spectra of electromagnetic waves by the magnetized plasma.** (**a**) Schematic diagram of the angle-resolved reflectivity detection of Weyl point. In this picture, 'Weyl point' is marked as red spot. Note that the azimuthal angle is positive when the incident plane lies between positive $x$ and $y$ axes. (the figure thus shows an angle with negative $\psi$) When the azimuthal angle $\psi$ is zero, the incident plane includes the projection of the Weyl point on the surface. (**b**) The degeneracy at the Weyl point as seen in the reflection spectrum as the pin point between the two unity reflectivity. (**c**,**d**) When the azimuthal angle is non-zero, the pin point opens up. (**e**,**f**) Intensity of reflected light from the angle-resolved reflection experiment at operating frequencies shifted from the plasmon Weyl point.

between the two bands (Fig. 3c,d). This clearly identifies that the degeneracy occurs at a single point in the $k$-space. Interestingly, EFSs of magnetized plasma can also be retrieved from the reflection intensity. As is shown in Fig. 3e,f, for $\omega_c = 1.2\omega_p$, at shifted frequencies from the plasmon Weyl points, the non-unity reflection regimes corresponds to the EFSs at each operating frequencies. Note that, unlike the solid-state example implicit in the lattice model of ref. 38, reaching this type I–type II transition requires no fine tuning in a magnetized plasma.

A remarkable phenomenon associated with the novel plasmon Weyl points is revealed by the polarization state of reflected light for a plane wave at $\omega = \omega_p$ incident at different angles. The Stokes parameter $S_3$ (ref. 39) and the polarization states of the reflected waves for both TE and TM incident waves are plotted in Fig. 4a,b, where arrows indicate the rotation direction of electric fields along with time evolution. For both TE and TM polarizations, the reflected wave's polarizations share the same feature that negligible polarization conversion happens in the negative $k_x$ region. However, the structure becomes quite complicated around the Weyl point when $k_x > 0$. The sudden change occurring at $k_x = 0$ can be explained by the abrupt change of the evanescent eigen-fields induced by the Weyl points. More explicitly, the evanescent 'eigen-states' (that is, with imaginary $k_y$) can be expressed as $[k_x, -\alpha k_z]^T$ for positive $k_x$, while $[0, 1]^T$ for the negative $k_x$. For negative Weyl point, however,

they are $[k_x, -\alpha k_z]^T$ for negative $k_x$, and $[0, 1]^T$ for positive $k_x$ (Supplementary Note 8 and Supplementary Fig. 8). Note that the basis vectors are the two degenerate states at the plasmon Weyl point—a circularly polarized propagating (that is, helical) mode and a longitudinal bulk plasmon mode. Thus, in the negative half $k$-plane ($k_x < 0$), the magnetized plasma shows bulk plasmon-like behaviour, whereas in the positive half $k$-plane ($k_x > 0$), the bulk plasmon is intermixed with the helical mode to generate a chiral response, which leads to a change in the polarization state. More remarkably, the half plane chirality could also be assigned to the polarization 'eigen-states' that diagonalize the reflection Jones matrices. As is shown in Fig. 4c, the 'eigen-states' are simply two orthogonal linear polarizations in the $k_x < 0$ half plane, while becomes very complex in the $k_x > 0$ half plane. When no loss is taken into account, the Jones matrices are unitary, and the 'eigen-states' are orthogonal with opposite-sign $S_3$ parameters and perpendicular orientation angles. $S_3$ parameter of the second 'eigen-state'(red) is shown as the colour in Fig. 4c. It could be shown that 'eigen-states' become circularly polarized when $\xi = k_z/k_x$ in the $k_x > 0$ half plane satisfy (Supplementary Note 9):

$$\xi = -\frac{1}{2n_s^2}\sqrt{\frac{\omega_p}{\omega_c}}\left(n_s^2 - \frac{\omega_c}{\omega_c - \omega_p}\right) \qquad (6)$$

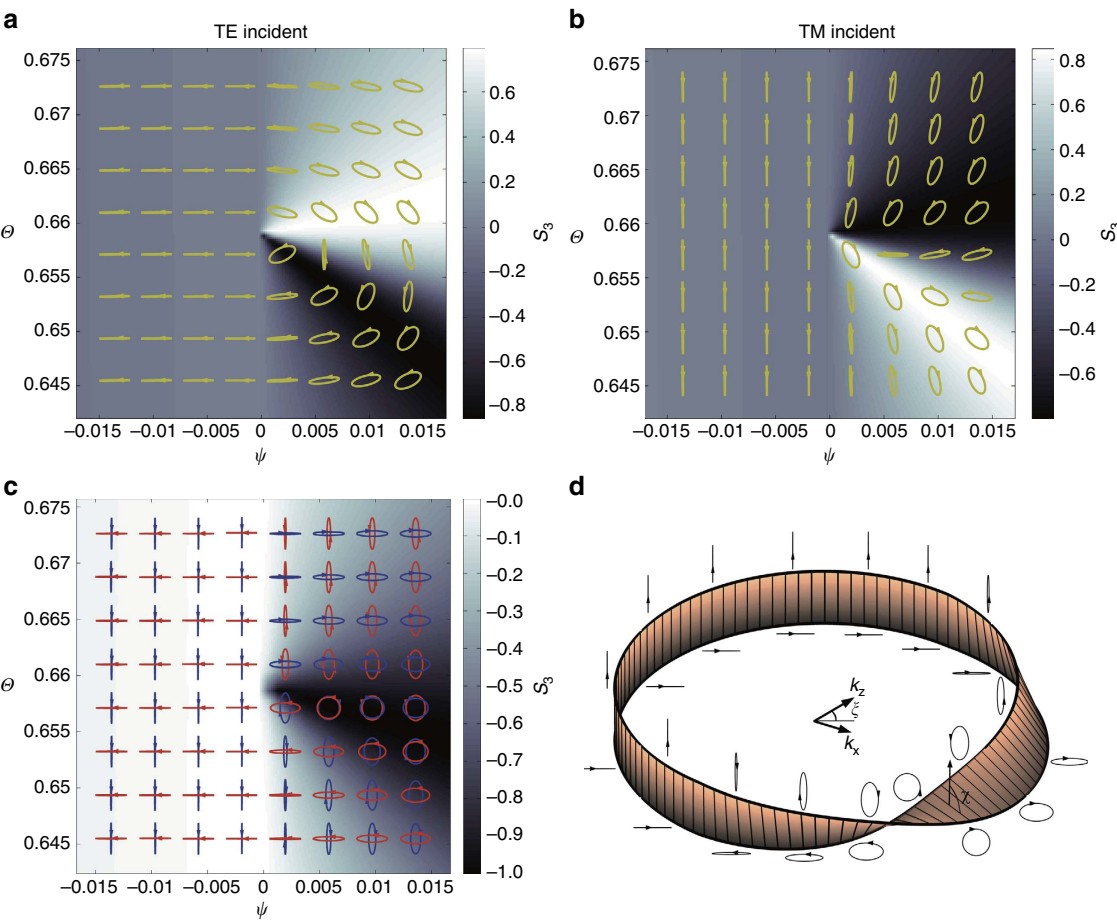

**Figure 4 | Polarization profiles of the reflected waves around the Weyl point. (a,b)** Polarizations and $S_3$ parameters (shown as the colour in background) of reflected light for TE polarization and TM polarization around the Weyl points (in the middle of the pictures). As is shown, when incident light is in negative $k_x$ regime, no apparent polarization conversion is observed, while $S_3$ parameters and polarizations change drastically in the positive regime. **(c)** Polarization eigen-states attained by diagonalizing the Jones matrices. In the $k_x > 0$ plane blue (red) loops represent right (left)-hand side elliptical polarization. $S_3$ parameters of the second eigen-state (red) is shown as background colour. **(d)** Mobius strip representation of the adiabatic evolution of polarization eigen-states around the Weyl points, the parameter loop needs to wind the Weyl point two times to get back to the initial states.

Thus, the eigen-states would continuously transit between linear and circular polarizations around the Weyl point, as is represented by the dark region in Fig. 4c. Intriguingly, eigen-states around the Weyl point satisfy a Mobius strip representation shown in Fig. 4d. We let the strip's edges represent the two eigen-states, and strip's rotation angle satisfy $\tan(\chi) = S_1/S_3$. When the eigen-state is circularly polarized, the strip is horizontal, while vertical for linear polarization. Exotic topology of the Mobius strip means that adiabatic evolution of eigen-states is required to wind the Weyl point two times to go back to the initial states, as is shown in Fig. 4d. Thus, the Weyl point is effectively an exceptional point for the polarization eigen-states.

Another important Weyl point feature is the 'Fermi arc' connecting Weyl points with opposite chiralities. When no loss is considered, 'Fermi arc' between Magnetized plasma and vacuum is shown in Fig. 5a ('Fermi arcs' at shifted frequencies are given in Supplementary Fig. 5). It could be derived that 'Fermi arc' in vicinity of the Weyl point is linear, and approach the Weyl point with slope that satisfy

$$\xi = -\frac{1}{4}\sqrt{\frac{\omega_p}{\omega_c}} - \frac{1}{2}\frac{\sqrt{\omega_p \omega_c}}{\omega_c - \omega_p} \qquad (7)$$

Presence of the 'Fermi arc' can also introduce intriguing polarization features seen by adding a vacuum layer (thickness $h$ in units of free space wavelength) sandwiched between the

magnetized plasma and a high refractive index medium in the reflection experiment. In practice, this can be achieved by a 'prism configuration' (Supplementary Fig. 9). In Fig. 5b, $|S_3|$ of polarization eigen-states under different $h$ are shown as functions of $\xi$. Apparently, as $h$ increases, transition of the eigen polarization states becomes sharper, for example, when $h = 0.1$, its polarization eigen-states are almost all linearly polarized except when near to the narrow dark region near the 'Fermi arc' (Fig. 5c). When $h$ is large enough, the polarization transition regime would converge to the 'Fermi arc' shown as blue line in Fig. 5c. This polarization transition region would correspond to a reflectance dip for locating 'Fermi arc' of this configuration when loss is introduced into the magnetized plasma. More intuitively, the sudden polarization transition across 'Fermi arc' could be viewed as sharper rotation in the Mobius strips representation, as is shown in Fig. 5d. 'Fermi arcs' could also be found between magnetized plasma and a perfect electric conductor (PEC) (Supplementary Fig. 6).

Possible experimental detection of the chirality of Weyl points has attracted considerable attentions in the solid-state sys-tems[40,41]. In our magnetized plasma system, since the chirality of polarization state occurs only in the half space, the chirality of Weyl point can be deduced by the side on which the chirality occurs. The chirality of the Weyl point is also reflected by the phase distribution of the reflection around the Weyl point. This spiralling phase around the Weyl point results in a vortex feature

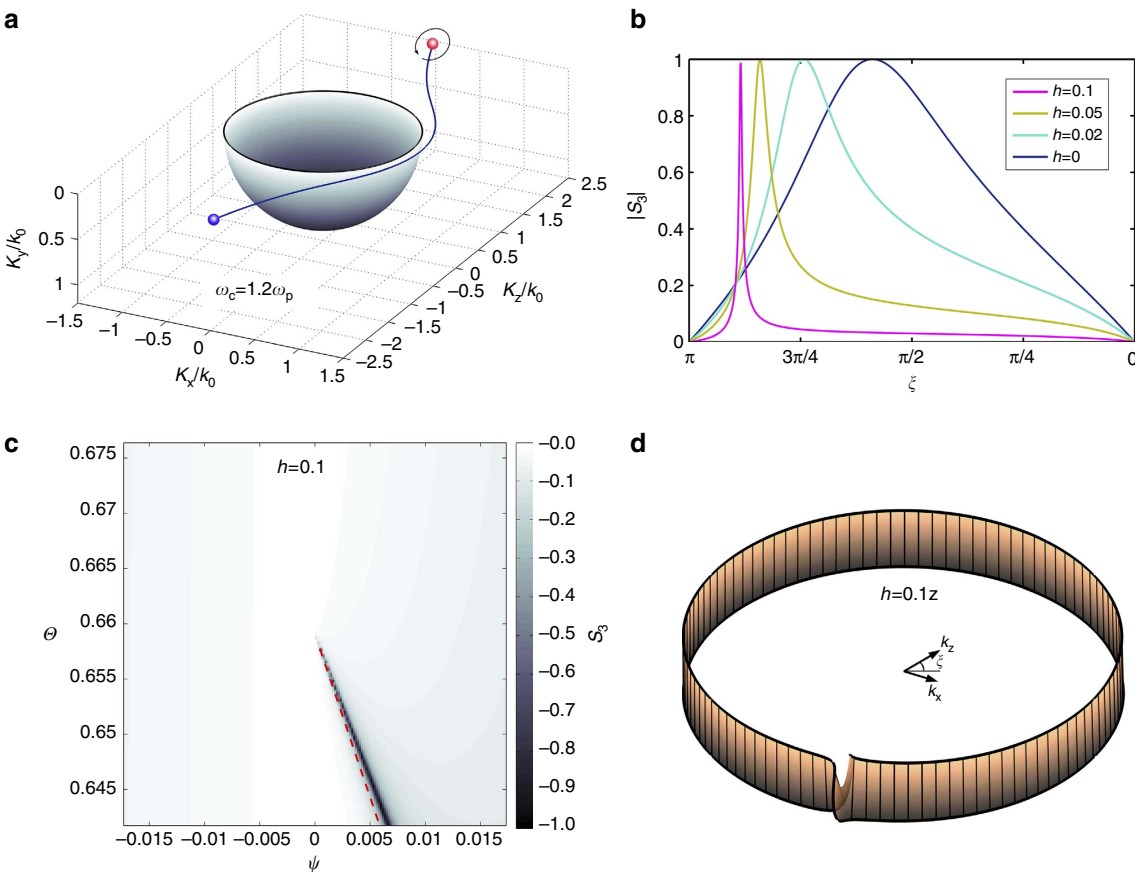

**Figure 5 | Polarization transition in presence of 'Fermi arcs'. (a)** Spatial dispersion of 'Fermi arc' between magnetized plasma and vacuum (shown as blue curve) connects Weyl points with opposite chiralities (red/blue for positive/negative Weyl point). The black loop indicates that we observe the polarization eigen-states around the positive Weyl point. **(b)** $|S_3|$ of polarization eigen-states show sharper transition feature as $h$ increases. The transition feature also gradually converges to the 'Fermi arc'. **(c)** 'Fermi arc' same in **a** is shown as red dashed line approaching the Weyl point. When $h = 0.1$, the polarization transition regime narrows down and gets already very close to the 'Fermi arc'. Further increase $h$ would make the regime finally converge to the 'Fermi arc'. **(d)** Mobius strip representation of the polarization eigen-states when $h = 0.1$. Different from the $h = 0$ case, the strip shows a very sharp turn, indicating a very sharp polarization eigen-state transition.

of the reflected beam when a TM-polarized Gaussian beam centred on Weyl points is incident onto the magnetized plasma. Furthermore, the chirality of the Weyl point is also indicated by the spin texture of the Fermi arc. The details of the above indications of the chirality of Weyl point are presented in Supplementary Note 10 and Supplementary Figs 10–13.

## Discussion

We finally discuss the presence of Weyl points in realistic plasma systems. Following the experimental work of ref. 42, we consider gaseous plasma with a density $N = 4.5 \times 10^{12}\,cm^{-3}$, the corresponding plasma frequency is given by $\omega_p = 1.2 \times 10^{11}\,rad\,s^{-1}$. According to equation (2), the presence of outer Weyl points requires $\omega_c > \omega_p$, corresponding to a minimum applied magnetic field of 0.68 T that can be readily achieved under laboratory conditions[43]. The Weyl points can also be observed in certain doped semiconductors such as indium antimonide (InSb) at terahertz frequencies. For an observed plasma frequency of 0.3 THz in an InSb sample[44,45], due to the small effective mass of electrons (0.014 $m_0$), the minimum strength of the applied magnetic field for generating outer Weyl points is only 0.15 T, which, again, is well within the reach of normal experimental conditions.

In conclusion, we report novel plasmon degeneracies in magnetized plasma, realizing the transition between type I and type II Weyl points. We have shown that these plasmon Weyl points lead to a number of striking optical signatures in terms of reflection intensities and polarization, and result in robust surface 'Fermi arcs'. With both magnetized plasmas and the required reflectometry readily available, our predictions may pave the way towards experiments in topological photonics with homogeneous dispersive systems and potential applications of surface Fermi arcs in constructing unidirectional waveguides.

**Data availability**. The data that support the findings of this study are available from the corresponding author on request.

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

## Acknowledgements

We thank Mike Gunn for stimulating discussions and feedback. This work was financially supported by ERC Consolidator Grant (Topological), the EPSRC (EP/J018473/1), Leverhulme Trust (RPG-2012-674) and the Opened Fund of the State Key Laboratory on Integrated Optoelectronics no. IOSKL2014KF12. B.B. and S.Z. acknowledge support from the Royal Society. F.F acknowledges support from National Natural Science Foundation of China (grant no.51320105009 and grant no.91423101)

## Author contributions

W.G., B.B. and S.Z. conceived the initial idea; W.G., B.Y., M.L., B.B. and S.Z. performed analytical modelling; W.G., M.L. and B.B. performed the surface state calculations; W.G.,

B.B. and S.Z. prepared the manuscript; S.Z. directed the project. All authors discussed the results and commented on the manuscript.

## Additional information

**Competing financial interests:** The authors declare no competing financial interests.

