## [Peer review file · Nature Communications]

Transferred manuscripts:

Reviewers' Comments:

Reviewer #1 (Remarks to the Author)

I am satisfied by the response and updates made by the authors. As I wrote in my first report, the manuscript is interesting and worth publishing. The revised manuscript is definitely improved and I recommend publication thereof in Nature Communications without further revisions.

Reviewer #2 (Remarks to the Author)

The authors largely improved the paper with additional data and detailed analysis. I highly recommend its publication in Nature Communications as is.

comments:

1) "semimetal" in the current title is highly inappropriate and confusing to different communities. A plasma system has no Fermi level.

Response to Reviewers

Reviewer #1:

Reviewer's comment:

I am satisfied by the response and updates made by the authors. As I wrote in my first report, the manuscript is interesting and worth publishing. The revised manuscript is definitely improved and I recommend publication thereof in Nature Communications without further revisions.

Our reply:

We thank the reviewer for recommending our paper for publication.

Reviewer #2:

Reviewer's comment:

The authors largely improved the paper with additional data and detailed analysis. I highly recommend its publication in Nature Communications as is.
comments:

1) "semimetal" in the current title is highly inappropriate and confusing to different communities. A plasma system has no Fermi level.

Our reply:

We thank the reviewer for the positive view of our work and for recommending our paper for publication. Based on the reviewer's suggestion, we change the title of our paper to "Photonic Weyl degeneracies in magnetized plasma".